# Silicon-containing water intake confers antioxidant effect, gastrointestinal protection, and gut microbiota modulation in the rodents

**Wei-Yi Wu[1], Pei-Li Chou[1], Jyh-Chin Yang[2]☯*, Chiang-Ting Chien[1]☯***

**1** Department of Life Science, School of Life Science, National Taiwan Normal University, Taipei, Taiwan, **2** Department of Internal Medicine, Hospital and College of Medicine, National Taiwan University, Taipei, Taiwan

☯ These authors contributed equally to this work.
* ctchien@ntnu.edu.tw (CTC); jcyang47@ntu.edu.tw (JCY)

**Data Availability Statement:** All relevant data are within the manuscript file.

**Funding:** This work was supported by grant from Ministry of Science and Technology in Taiwan

## Abstract

We explored the effects of silicon-containing water (BT) intake on gastrointestinal function and gut microbiota. BT was obtained by pressuring tap water through silicon minerals (mullite, $Al_6Si_2O_{13}$) column. BT decreased $H_2O_2$ chemiluminescence counts, indicating its antioxidant activity. Four weeks of BT drinking increased $H_2O_2$ scavenging activity and glutathione peroxidase activity of plasma. BT drinking did not affect the body weight but significantly reduced the weight of feces and gastrointestinal motility. BT drinking significantly suppressed pylorus ligation enhanced gastric juice secretion, gastric reactive oxygen species amount, erythrocyte extravasation, IL-1β production by infiltrating leukocyte, and lipid peroxidation within gastric mucosa. Data from 16S rRNA sequencing revealed BT drinking significantly increased beneficial flora including *Ruminococcaceae* UCG-005, *Prevotellaceae* NK3B31, *Weissella paramesenteroides*, *Lactobacillus reuteri*, and *Lactobacillus murinus* and decreased harmful flora including *Mucispirillum*, *Rodentibacter*, and *Staphylococcus aureus*. This study pioneerly provided scientific evidences for the potential effects of water-soluble forms of silicon intake on antioxidant activity, gastrointestinal function, and gut microbiota modulation.

## Introduction

Silicon is an essential micronutrient and is the third most abundant trace element in the human body. The major food sources of silicon are cereals, oats, barley, white wheat flour, polished rice, mineral water, and beer. In contrast, animal source foods such as meat and dairy products contain lower level of silicon. Silicon is naturally present in food as several forms of silicate, including silicon dioxide ($SiO_2$), free ortho-silicic acid ($H_4SiO_4$), or silicic acids bounded to certain nutrients [1].

Mounting evidences indicate that silicon possesses pharmacological effects and plays an essential role in health including bone mineralization, collagen synthesis, aging of skin, integrity of hair and nails, atherosclerosis, and other disorders [2]. For example, dietary silicon

(MOST-107-2218-E-003-001) and Bestec Biotechnology Co., Ltd. in Taiwan. Ministry of Science and Technology provided financial support in the form of salaries for CTC and WYW and research material. Bestec Biotechnology Co., Ltd. provided financial support in research material. The specific roles of these authors are articulated in the 'author contributions' section. The funders had no role in study design, data collection and analysis, decision to publish, or preparation of the manuscript.

deficiency researches on laboratory animals showed stunted growth and profound defects in bone and other connective tissues, which suggested the role for silicon in normal growth and development in higher animals [3]. In addition, some studies stated that higher dietary silicon intake was correlated with higher bone mineral density in men, pre-menopausal women, and postmenopausal women taking hormone replacement therapy (HRT) but not in postmenopausal women not taking HRT [4].

Previous research revealed that organic silicon possessed the antioxidant, anti-apoptotic, and neuroprotective potential against hydrogen peroxide toxicity in human neuroblastoma cell line [5]. In addition, the beneficial effect of silicon incorporated in a restructured pork matrix has been found in aged rats fed with high-saturated fat and high-cholesterol diet. Dietary enrichment with silicon enhanced hepatocyte antioxidant defenses by removing hydrogen peroxide-induced oxidative stress [6]. These evidences informed that silicon may participate in antioxidant defense and anti-apoptosis pathway, and may be a prominent therapeutic substance in humans.

The composition of the intestinal microbiota varies among individuals and throughout lifetime, due to various environmental and genetic factors. Appropriate composition of gut microbiota is essential for hosts to maintain gastrointestinal tract health [7]. However, disturbance of gut microbiota is correlated with damaged intestinal epithelium and other gut barrier dysfunction, which allows for lipopolysaccharide penetration and causes metabolic endotoxemia [8, 9]. Several kinds of intestinal flora, such as *Escherichia coli*, *Enterobacteriaceae*, and *Bacteroides*, are considered as biomarkers in inflammatory bowel diseases [10]. These pathogenic bacteria can destroy intestinal barrier and subsequently invade the body leading to endotoxemia [11]. In contrast, several probiotics and their metabolites may depress the proliferation of pathogenic bacteria, and increase the intestinal barrier function [12, 13]. Previous research using administration of colloidal silicon dioxide in tablet form in patients with acute diarrhea revealed that the high-dispersion silicon dioxide enterosorbent exhibited antidiarrheal effect [14]. Moreover, it is reported that silicon dioxide blocks the receptors of mucous membrane that is responsible for pathogens adhesion and toxins binding, and accelerates the adsorption of active substances in intestine [15]. Although several biological effects of dietary silicon have been explored, its influences on gastrointestinal function and intestinal microbiota composition have not been fully determined.

In this study, we designed a setup to produce silicon-containing water (BT) by guiding tap water via a pressure gradient through one activated carbon column, two ion exchange resin columns, one activated filter column, one silicon minerals (mullite, $Al_6Si_2O_{13}$) column, one UV sterilizer and magnetization. This setup can recirculate the water with silicon minerals and produce BT with different $SiO_2$ concentration. We aimed to explore the effects of drinking silicon-containing water (BT) on human health or disease prevention through rodent models. Concisely, this preclinical research was designed to evaluate the characteristics of silicon-containing water and its multifaceted influences on reactive oxygen species scavenging activity, gastrointestinal function, and gut microbiota.

## Methods and materials

### A setup for BT preparation

BT was provided by Bestec Biotechnology Company (Bestec Biotechnology Co., LTD, Taipei, Taiwan). Before preparing the BT, tap water was guided via a pressure gradient through one activated carbon column, two ion exchange resin columns, one activated filter column, one silicon-mineral (mullite, $Al_6Si_2O_{13}$) column, one UV sterilizer, and magnetization, respectively, as shown in **Fig 1A**. With increased time of recirculation preparation procedure, the different concentration of BT

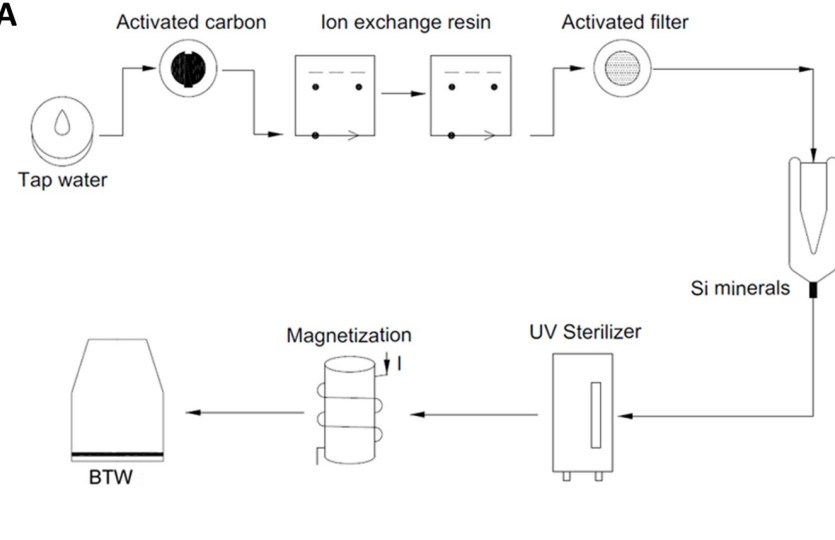

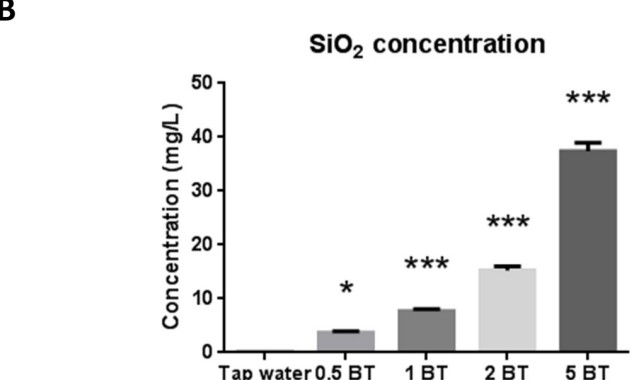

**Fig 1. Setup of BT preparation and characteristic of BT.** (A) A setup for BT preparation. BT was produced by filtering tap water through one activated carbon column, two ion exchange resin columns, one activated filter column, one Si minerals column, one UV sterilizer and magnetization. (B) $SiO_2$ concentration of BT. With increased time of recirculation preparation procedure, the higher concentration of BT containing $SiO_2$ was produced. The $SiO_2$ concentration in the prepared BT was time-dependently increased in the 0.5BT, 1BT, 2BT, and 5BT as compared with tap water (n = 3). 0.5 BT, 12 h of recirculation preparation; 1 BT, 24 h of recirculation preparation; 2BT, 48 h of recirculation preparation; 5 BT, 120 h of recirculation preparation. The data are mean ± SEM and analyzed by one-way ANOVA. $^{*}P < 0.05$; $^{***}P < 0.001$.

was obtained. Accordingly, we produced 0.5BT (12 hour recirculation), 1BT (24 hour recirculation), 2BT (48 hour recirculation), and 5BT (120 hour recirculation) in sequence.

## Assay of water $SiO_2$ concentration

The dissolved silica concentration was determined by molybdosilicate method (APHA 1998). Standard solution of different concentration ranging from 10 to 40 ppm was prepared by dissolving $Na_2SiO_3$. The sample were stirred well and kept for three hours to complete the reaction. The optical density was measured for standards and water sample at 812 nm using UV-visible spectrophotometer (CT-3800, ChromTech, Kingtech Scientific Co., Ltd., Taipei, Taiwan).

## Animals

Forty male 7-week-old Wistar rats and fifty male 7-week-old C57BL/6 mice were purchased from BioLASCO Taiwan Co., Ltd, (Yi-Lan, Taiwan). Animals were housed in the animal

center of National Taiwan Normal University at a controlled room temperature under a 12 hours dark-light cycle with free access to food and tap water. After one week period of accommodation, rats and mice were divided in to five groups respectively, eight rats per group and ten mice per group, including control (tap water), 0.5 BT, 1 BT, 2 BT, and 5 BT drinking. All treatments were used as daily drinking water for 4 weeks. Animal body weights were measured before and after treatments weekly. All animal experiments were performed in accordance with the guidelines of the National Science Council of the Republic of China (1997). This study was approved by the Institutional Animal Care and Use Committee of National Taiwan Normal University (No. 107015). All animal experiments were performed under anesthesia, and all efforts were made to minimize suffering.

## Assay of gastrointestinal propulsion

In order to evaluate the effects of drinking silicon-containing water on gastrointestinal motility, we utilized charcoal meal test for the measurement of gastrointestinal transit [16]. After four weeks of experimental treatments, mice were housed in cage individually with free access to food and water. Fecal samples of mice were collected in a period of 24-hours for the measurement of daily fecal weight. Afterwards, all mice were deprived of food for 24 hours but with free access to water before experimentation. Gastrointestinal transit of mice was evaluated by the transport of a test meal containing non-absorbable marker, charcoal (Sigma-Aldrich, St. Louis, USA). In brief, the test meal (0.1 ml) was administered intragastrically by oral gavage feeding tube. The mice were anesthetized with intraperitoneal injection of urethane (1.2 g/kg, Sigma-Aldrich, St. Louis, USA) thirty minutes after test meal administration. After the small intestines of mice were harvested rapidly by laparotomy, mice were sacrificed by intravenous injection of potassium chloride. Gastrointestinal transit was shown as the percentage of the length of the small intestine traversed by the charcoal marker divided by the total length of the small intestine.

## Assay of gastroprotective effect of BT

In this part of study, pylorus ligation is a typical method for the uniform production of gastric ulceration in rat, thereby being commonly used in the assessment of antiulcer substance [17]. After four weeks of experimental treatments, rats were deprived of food for 24 hours but with free access to water before experimentation. After rat was anesthetized with intraperitoneal injection of urethane, the stomach was exposed by midline laparotomy, and the pylorus ligation was conducted. Four hours after the pylorus ligation, the stomach was harvested, and the gastric juice was collected into graduated test tube.

## *In vitro* chemiluminescence recording for reactive oxygen species

The free radical level of the stomach tissue was measured by luminol chemiluminescence detection method [18]. Concisely, a piece of freshly harvested stomach tissue from rat was mixed with 0.5 ml of 0.1 mmol/L luminol (5-amino-2,3-dihydro-1,4-phthalazinedione, Sigma, Chemical Co., St. Louis, USA) and was analyzed with a chemiluminescence analyzing system (CLD-110, Tohoku Electronic Inc. Co., Sendai, Japan). Firstly, the detection were conducted on stomach tissues without luminal for 60 seconds, which were set as baseline. Afterwards, the chemiluminescence signals emitted from the mix of stomach tissue and luminol, which represented the hydrogen peroxide content in the stomach lumen, were recorded for 240 seconds. In addition, we evaluated the free radical scavenging activity of several dosages of BT and rat plasma from each groups. Briefly, 0.2 ml of test samples were mixed with 0.5 ml of luminol and 0.1 ml of $H_2O_2$ (0.03%, Sigma-Aldrich, St. Louis, USA). Similarly, the detection were

performed on test samples for 60 seconds, which were set as baseline. Subsequently, the test samples were added with luminal and $H_2O_2$. The enhanced chemiluminescent signals from the sample-luminol-$H_2O_2$ mixure were recorded for 180 seconds. The chemiluminescent (CL) counts were measured every 10 seconds. The total CL counts representing the hydrogen peroxide levels count in luminal detection method were calculated from the sum of each CL counts with baseline correction.

## Glutathione peroxidase activity assay

Glutathione peroxidase (GPx) activity of rat plasma from each group were analyzed with glutathione peroxidase assay kit (Cayman Chemical Company, Ann Arbor, MI, USA). The oxidized glutathione, produced upon the reduction of hydroperoxide by GPx, is recycled to glutathione by glutathione disulfide reductase and nicotinamide adenine dinucleotide phosphate (NADPH). The oxidation of NADPH to $NADP^+$ is accompanied by a decrease in the absorbance at 340 nm. The rate of decrease in the $A_{340}$ is directly proportional to the GPx activity in the plasma, which was expressed in nmol/min/ml.

## Histological examination

The stomach tissue were fixed with 10% formalin in phosphate-buffered saline for 24-hours and embedded in paraffin. Sections (5 μm) of rat stomach were sliced with microtome (RM 2125 RTS, LEICA, Germany). Sections were mounted on slides, and stained with hematoxylin and eosin (H&E) for pathological examinations. We utilized light microscopic evaluation to analyze the histopathological changes of the stomach including erythrocyte extravasation and leukocyte infiltration of stomach mucosa. We performed immunohistochemistry to evaluate the oxidative damage and the pro-inflammatory response of stomach mucosa in each group. In brief, sections of rat stomach were deparaffinized, rehydrated, and immunohistochemically stained with 4-hydroxynonenal (4-HNE) (1:400; BIOSS, Boston, MA, USA), 3-nitrotyrosine (3-NT) (1:400; Abcam, Cambridge, MA, USA), and interleukin-1 beta (IL-1β) (1:500; Abcam, Cambridge, MA, USA) antibodies, respectively. Quantitative analysis of histological examination was performed with software ImageJ (National Institutes of Health, Bethesda, MD, USA). Three sections from each group were selected for the quantification of erythrocyte extravasation, 4-HNE, 3-NT, and IL-1β staining through analyzing the area of erythrocyte and the brown signals of 4-HNE, 3-NT, and IL-1β divided by the total area of the tissue.

## DNA extraction and sequencing of gut microbiota

Fecal samples were collected from rats after 4 weeks of experimental treatments. Fecal microbiota DNA was extracted using the QIAamp DNA Stool Mini Kit (Qiagen, USA). The next-generation sequencing of bacterial 16 S ribosomal RNA genes following previous procedure were conducted to distinguish the intestinal bacteria [19]. The V3–V4 regions of 16S rRNA genes, which were generally used for intestinal microbiome studies, were amplified using a specific primer with a barcode. Fecal microbiota composition was assessed using Illumina HiSeq sequencing of 16S rDNA amplicon and QIIME-based microbiota analysis. Operational taxonomic unit (OTU) clustering and species annotation were performed from representative sequences using UPARSE software (Version 7.0.1001) and the Greengenes Database based on Ribosomal Database Project classifier (Version 2.2), respectively. OTUs abundance information was normalized using a standard of sequence number corresponding to the sample with the least sequences.

## Statistical analysis

GraphPad Prism 6 (GraphPad Software Inc., CA, USA) was used for graphing and statistical analysis. All values were expressed as the mean ± standard error of the mean (SEM). All parameters were compared by one-way analysis of variance (ANOVA) to assess differences among groups. Differences within each groups were analyzed by Student's paired t-test. $P < 0.05$ indicated a statistical significance.

## Results

### SiO$_2$ concentration of BT

The SiO$_2$ concentration of all types of BT was demonstrated in **Fig 1B**. Our data showed that SiO$_2$ concentration was time-dependently increased in the 0.5BT (3.740 ± 0.187 vs. 0.106 ± 0.005, P = 0.0153), 1BT (7.732 ± 0.187 vs. 0.106 ± 0.005, P<0.0001), 2BT (15.280 ± 0.729 vs. 0.106 ± 0.005, P<0.0001), and 5BT (37.400 ± 1.556 vs. 0.106 ± 0.005, P<0.0001) water samples as compared with tap water. This data informed that a BT setup can stably produce SiO$_2$ containing water.

### Hydrogen peroxide scavenging activity of BT

We compared the hydrogen peroxide (H$_2$O$_2$) scavenging activity of RO water and four concentrations of BT. Our data showed that 0.5BT (120026 ± 40147 vs. 416237 ± 131295, P = 0.0237), 1BT (59131 ± 24395 vs. 416237 ± 131295, P = 0.0079), 2BT (49762 ± 7524 vs. 416237 ± 131295, P = 0.0067), and 5BT (47075 ± 7890 vs. 416237 ± 131295, P = 0.0064) significantly reduced H$_2$O$_2$-induced chemiluminescent counts, which demonstrated BT possessed H$_2$O$_2$ scavenging activity as compared with RO water (**Fig 2A and 2B**).

### Hydrogen peroxide scavenging activity of plasma

Four weeks after drinking all kinds of water, we compared the H$_2$O$_2$ scavenging activity of rat plasma from each goup. Our data revealed that plasma from 5 BT drinking group (26753 ± 1187 vs. 56240 ± 12369, P = 0.0444) significantly reduced H$_2$O$_2$-induced chemiluminescent counts, as compared with plasma from tap water drinking group (**Fig 2C and 2D**). This result indicated that drinking BT can elevate antioxidant capacity of plasma.

### Glutathione peroxidase activity of plasma

We analyzed the glutathione peroxidase activity (GPx) of rat plasma from each group after four weeks of different water drinking treatments. Our result demonstrated that 5 BT drinking (63.62 ± 6.189 vs. 49.05 ± 1.266, P = 0.0412) significantly increased GPx activity of plasma, as compared with tap water drinking group (**Fig 2E**).

### Status of animals and defecation recording

We recorded the basal physiological parameters including the body weight, dry weight of feces and the ratio of dry weight of feces to body weight in five groups of mice. As shown in **Fig 3A**, the increase of body weight was similar in all groups of animals. However, the dry weight of feces (**Fig 3B**) and the ratio of dry weight of feces to body weight (**Fig 3C**) were significantly decreased in the 0.5BT (1.243 ± 0.096 vs. 2.263 ± 0.211, P<0.0001), 1BT (1.487 ± 0.089 vs. 2.263 ± 0.211, P = 0.0007), 2BT (1.375 ± 0.122 vs. 2.263 ± 0.211, P<0.0001), and 5BT (1.222 ± 0.083 vs. 2.263 ± 0.211, P<0.0001) groups as compared with

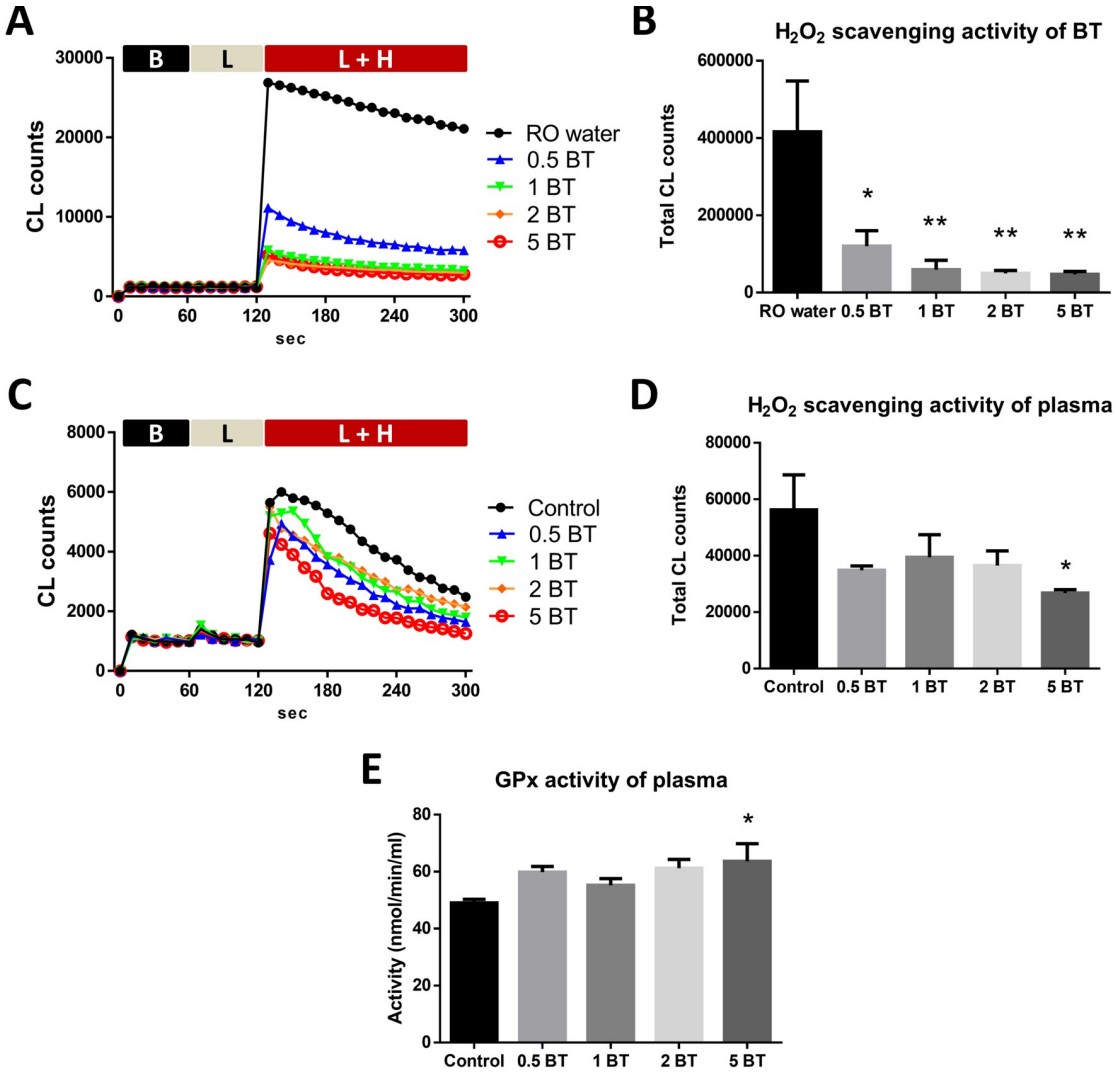

**Fig 2. Antioxidant activity of BT and plasma.** (A) The curves of CL counts in several kinds of water. B, baseline; L, luminal; H, $H_2O_2$. (B) $H_2O_2$ scavenging activity of BT. $H_2O_2$ scavenging activity was indicated by CL counts in several kinds of water. CL counts were decreased by 0.5 BT, 1 BT, 2 BT and 5 BT (n = 3). RO, reverse osmosis. (C) The curves of CL counts in plasma of different groups. B, baseline; L, luminal; H, $H_2O_2$. (D) $H_2O_2$ scavenging activity of plasma. CL counts were decreased by plasma of 5 BT drinking group (n = 3). (E) Glutathione peroxidase activity of plasma. The plasma from 5 BT drinking group demonstrated increased glutathione peroxidase activity (n = 3). GPx, glutathione peroxidase. The data are mean ± SEM and analyzed by one-way ANOVA. *$P < 0.05$; **$P < 0.01$.

control group. Moreover, there was no significant difference of moisture content of the feces among all groups (**Fig 3D**).

## Effects of BT on gastrointestinal motility

We determined the gastrointestinal motility with the charcoal meal test on mice, which displayed the gastrointestinal transit rate of the small intestine in the rodents. As shown in **Fig 4**, there were no significant effects for BT drinking on gastrointestinal transit rate in the 0.5BT, 1BT, and 2BT groups. However, 5BT drinking group (64.750 ± 4.390 vs. 90.090 ± 5.168, P = 0.0005) displayed a significantly reduced gastrointestinal transit rate as compared with control group, which represented a depressed small intestinal motility.

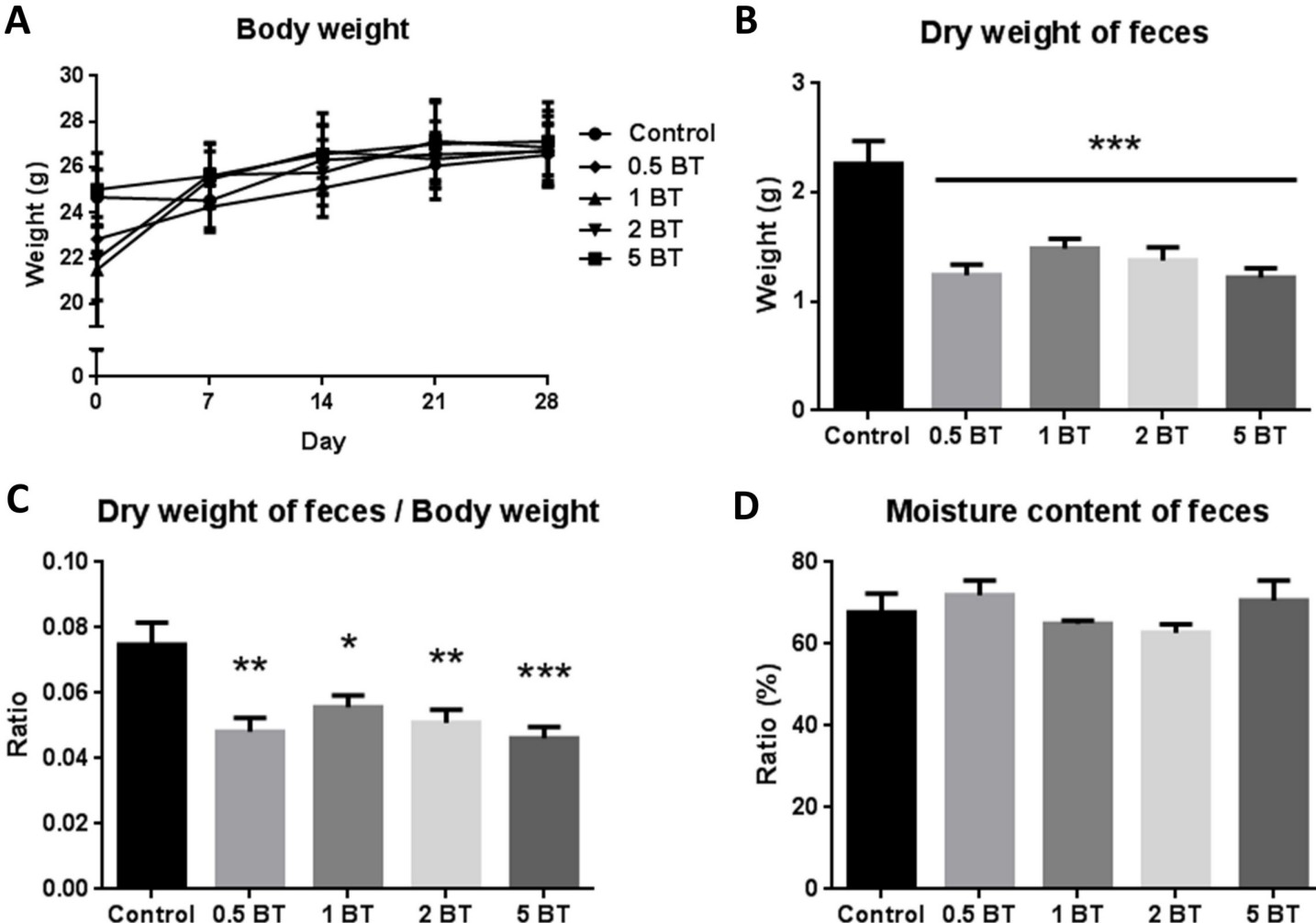

**Fig 3. Status of animals and defecation recording.** The effects of drinking BT on the physiological parameters of body weight (A), dry weight of feces (B), the ratio of dry weight of feces to body weight (C), and moisture content of feces (D) in these five groups of mice (n = 10). All types of BT decreased dry weight of feces. The data are mean ± SEM and analyzed by one-way ANOVA. $^*P$ <0.05; $^{**}P$<0.01; $^{***}P$<0.001.

### Effects of BT on pylorus ligation induced ulcer

Because pylorus ligation stimulated the gastric juice secretion in vagally intact rats, we determined the volume of gastric juice response to pylorus ligation in these five groups of rats. As shown in **Fig 5A**, our result indicated that only the 5BT group (0.750 ± 0.130 vs. 1.885 ± 0.309, P = 0.0361) displayed a significant decreased gastric juice secretion as compared with control group. Pylorus ligation increased the reactive oxygen species (ROS) amount of gastric lumen in the control, 0.5BT, 1BT, 2BT groups as compared with normal group (Normal group, naïve rats without any experimental treatment) (**Fig 5B and 5C**). However, a significant depressed ROS production was noted in the 5BT group (311 ± 259.2 vs. 2262 ± 814.6, P = 0.0470) as compared with control group. The histopathological evaluation of gastric mucosa indicated the increase of erythrocyte extravasation, leukocyte infiltration and oxidative injury after pylorus ligation with the excess accumulation of gastric juice (**Fig 5D**). As shown in **Fig 5E**, quantitative analysis revealed the significantly decreased percentage of the area of erythrocyte extravasation in 5 BT group (0.4310 ± 0.1018 vs. 2.402 ± 0.7618, P = 0.0084), lipid peroxidation product (4-HNE)

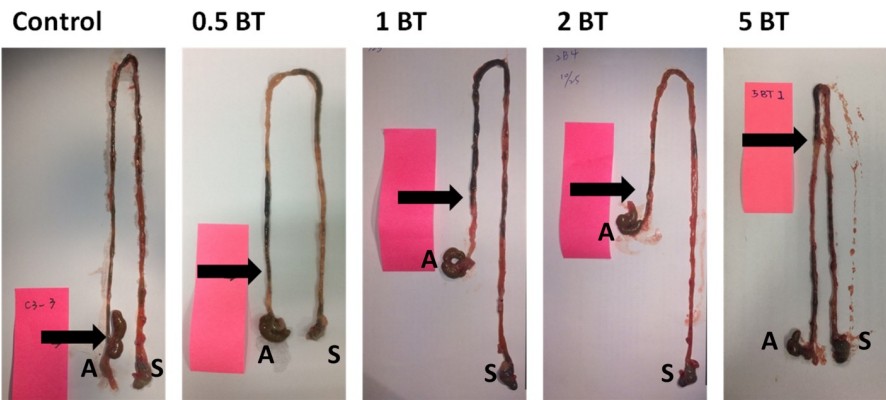

## Gastrointestinal transit

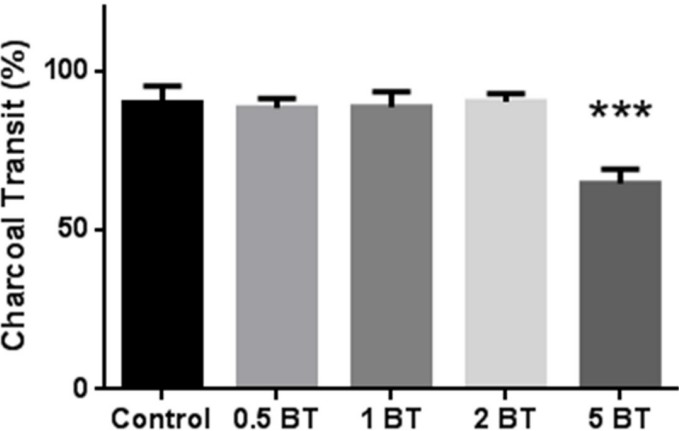

**Fig 4. Effects of BT on gastrointestinal motility.** The effects of drinking BT on gastrointestinal transit in the five groups of mice (n = 10). Black arrows indicate the intestine traversed by the charcoal marker. S, Stomach; A, appendix. Only 5 BT decreased gastrointestinal transit. The data are mean ± SEM and analyzed by one-way ANOVA. ***$P$ <0.001.

within gastric mucosa in 2 BT (4.709 ± 0.3107 vs. 7.741 ± 0.5362, P = 0.0104) and 5 BT groups (4.008 ± 0.3125 vs. 7.741 ± 0.5362, P = 0.0023), and IL-1β, which was produced by infiltrating leukocyte, in 5 BT group (3.000 ± 0.4775 vs. 6.355 ± 0.5529, P = 0.0442), as compared with control group. However, there were no obvious variations of tyrosine nitration (3-NT) mediated by reactive nitrogen species within gastric mucosa among different water drinking groups. These results indicated that drinking BT may reduce gastric mucosal damage through antioxidant activity, which were corresponded to the elevated systemic $H_2O_2$ scavenging activity.

## Variations of gut microbiota after BT treatment

We performed 16S rRNA gene sequencing with the collected feces from rats to determine the gut microbiota composition and analyzed the relative abundance of specific bacterial taxa to clarify the effect of BT in gut microbiota alteration. The results of the histogram of genus (Fig 6A), which represented the relative abundance of the top 10 most abundant genera of each groups, revealed a few variations among groups. We observed that 0.5BT (0.02238 ± 0.002700 vs. 0.007793 ± 0.001514, P = 0.0089) and 2BT (0.02090 ± 0.003414 vs. 0.007793 ± 0.001514, P = 0.0207) groups had significantly increased relative abundance of *Ruminococcaceae* UCG_005.

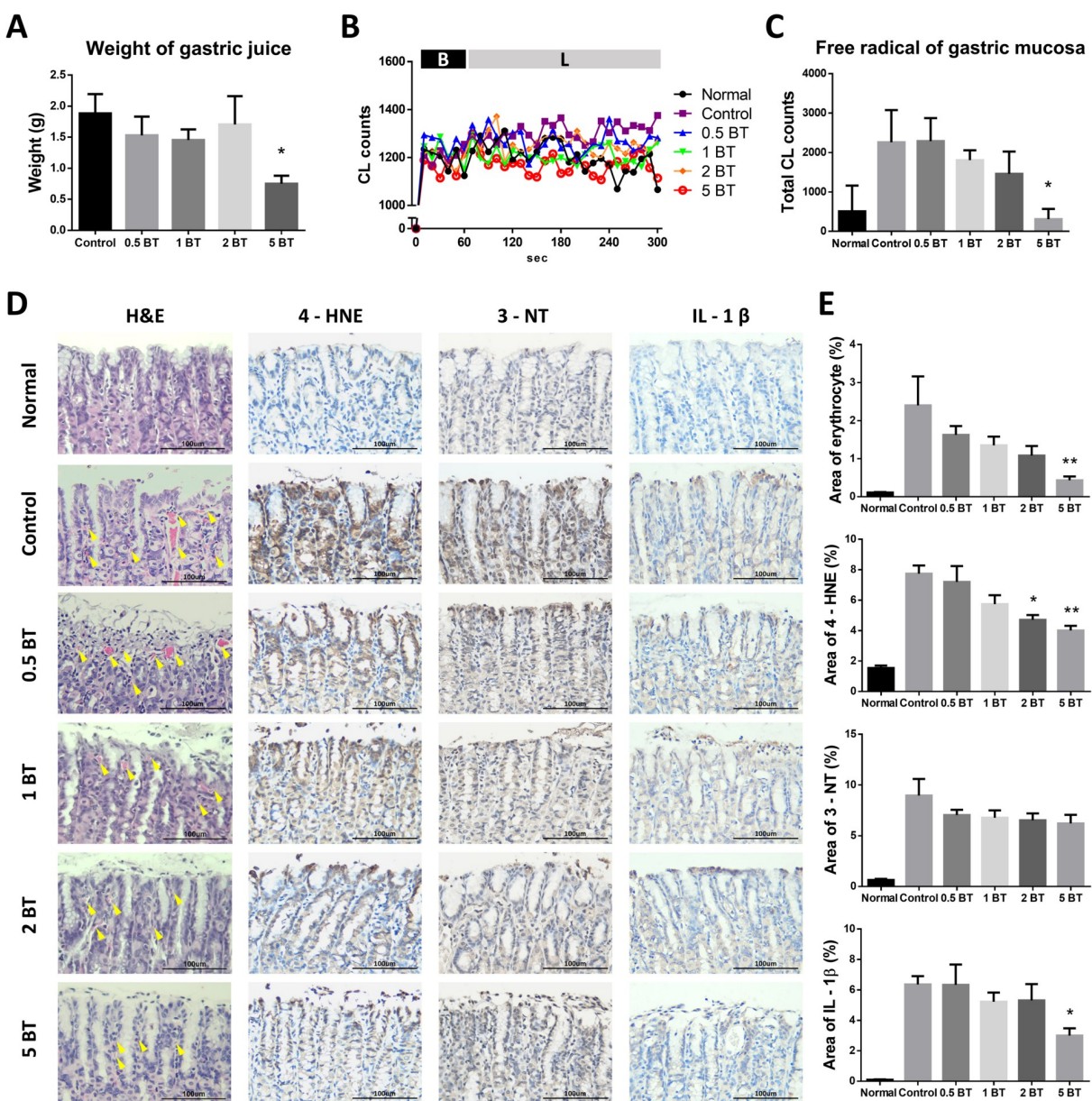

**Fig 5. Effects of BT on pylorus ligation induced ulcer.** The effects of drinking BT on pyloric ligation induced gastric juice secretion (A), the curves of CL counts from gastric mucosa free radicals content (B), and the free radicals content within gastric mucosa (C). B, baseline; L, luminol. (D) The histopathological evaluation of gastric mucosa. Yellow arrows indicate erythrocyte extravasation. (E) Quantitative analysis of histopathology. The area of different targets were shown as percentage (%). Gastric juice, gastric mucosa free radicals, erythrocyte extravasation, and area of IL-1β were decreased in 5 BT group. Area of 4-HNE were decreased in 2 BT and 5 BT groups (n = 8). The data are mean ± SEM and analyzed by one-way ANOVA. *$P < 0.05$; **$P < 0.01$.

Relative abundance of *Prevotellaceae* NK3B31_group was significantly increased in 1BT (0.08718 ± 0.01854 vs. 0.02202 ± 0.02209, P = 0.0272) and 2BT (0.1123 ± 0.02256 vs. 0.02202 ± 0.02209, P = 0.0015) groups. Relative abundance of *Prevotellaceae* UCG_001 was significantly decreased in 0.5BT (0 vs. 0.04730 ± 0.01355, P = 0.0032) and 1BT (0.01013 ± 0.003838 vs. 0.04730 ± 0.01355, P = 0.0246) groups. Apart from the top 10 most abundant genera, we further analyzed a few relative abundances at genus level among groups (**Fig 6B**). Relative abundance of *Mucispirillum* was significantly decreased in the 0.5BT (0.0001274 ± 0.0001178 vs. 0.0005215 ±

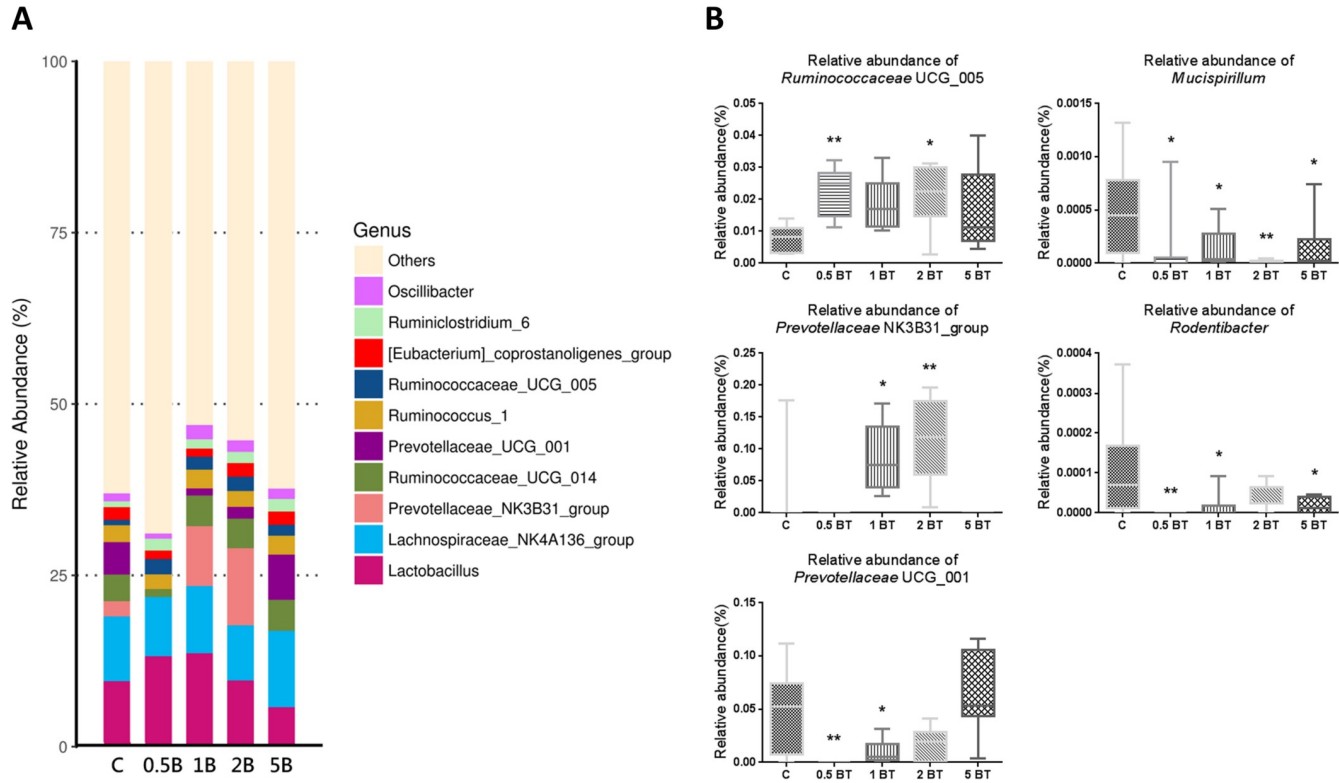

**Fig 6. Variations of gut microbiota after BT treatment at genus level.** (A) Effects of drinking BT on the relative abundance of the top 10 most abundant flora of gut microbiota at genus level. Bacteria represented by different colors are marked in the upper right of Fig 6A. (B) Drinking BT increased the relative abundance of *Ruminococcaceae* UCG_005 and *Prevotellaceae* NK3B31_group, whereas decreased the relative abundance of *Prevotellaceae* UCG_001, *Mucispirillum*, and *Rodentibacter* (n = 8). C, control; 0.5B, 0.5BT; 1B, 1BT; 2B, 2BT; 5B, 5BT. Data of relative abundance are depicted in box and whisker plots and are analyzed by one-way ANOVA. $^*P$ <0.05; $^{**}P$<0.01.

0.0001547, P = 0.0304), 1BT (0.0001361 ± 0.0000654 vs. 0.0005215 ± 0.0001547, P = 0.0352), 2BT (0.0000116 ± 0.0000062 vs. 0.0005215 ± 0.0001547, P = 0.0037), and 5BT (0.0001420 ± 0.00009196 vs. 0.0005215 ± 0.0001547, P = 0.0388) groups. Relative abundance of *Rodentibacter* was significantly decreased in 1BT (0.00001449 ± 0.00001154 vs. 0.0001072 ± 0.00004335, P = 0.0122) and 5 BT (0.0000174 ± 0.000007271 vs. 0.0001072 ± 0.00004335, P = 0.0156) groups and was not detected in 0.5 BT (0 vs. 0.0001072 ± 0.00004335, P = 0.0033) group.

Linear discriminant analysis effect size (LEfSe) (**Fig 7A**) was calculated to determine the specific bacterial taxa that were predominant among groups. We further analyzed the relative abundance at species level among groups (**Fig 7B**). Relative abundance of *Weissella parame-senteroides* was significantly increased in 2BT (0.0001073 ± 0.00003795 vs. 0.0000058 ± 0.000058, P = 0.0026) group. Relative abundance of *Lactobacillus reuteri* was significantly increased in 0.5BT (0.04161 ± 0.01008 vs. 0.01714 ± 0.003338, P = 0.0281) group. Relative abundance of *Lactobacillus murinus* was significantly increased in 1BT (0.05977 ± 0.01867 vs. 0.01075 ± 0.006191, P = 0.007) group. Furthermore, *Staphylococcus aureus* was not detected in 5BT group, whereas there was no significant difference between control group and 5BT group (P = 0.5826).

## Discussion

Emerging scientific evidences suggest the benefit from consumption of water-soluble forms of silicon, which indicate a potential beneficial effect of silicon on human health [20]. It has been

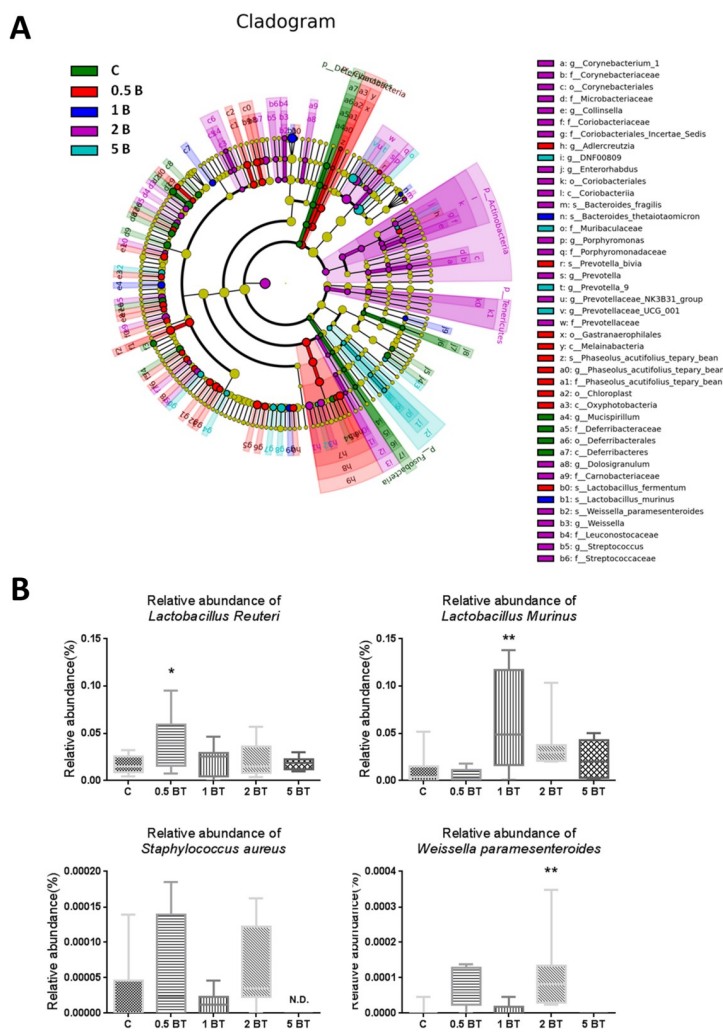

**Fig 7. Variations of gut microbiota after BT treatment at species level.** (A) LEfSe taxonomic cladogram derived from 16S sequences of gut microbiota among groups. Different colors represented significantly different taxa among groups. (B) The effects of drinking BT on the relative abundance of gut microbiota in species level. Drinking BT increased the relative abundance of *Weissella paramesenteroides*, *Lactobacillus reuteri*, and *Lactobacillus murinus*, whereas decreased the relative abundance of *Staphylococcus aureus* (n = 8). C, control; 0.5B, 0.5BT; 1B, 1BT; 2B, 2BT; 5B, 5BT; N.D., not detected. Data of relative abundance are depicted in box and whisker plots and are analyzed by one-way ANOVA. $^*P < 0.05$; $^{**}P < 0.01$.

revealed that silicon may influence the metabolism of different organs and possess several biological effects such as bone mineralization, collagen synthesis, prevention of aging, modulation of antioxidative enzymes, reduced risk of atherosclerosis and nervous system diseases [20, 21]. However, detailed researches on the biological roles of silicon in gastrointestinal function are still lacking.

Exacerbated reactive oxygen species (ROS) production causes several types of damage to cellular structures, resulting in the development of a variety of gastrointestinal diseases such as gastric ulcer and intestinal ulcer. An advanced technology has been established for generation of reduced water with a lower oxidation-reduction potential (ORP) and antioxidant activity [22]. The low ORP or reduced water can be obtained by activating the water using electrolysis, magnetic field, collision, or minerals, which is similar to our BT water. In the present study,

our data of *in vitro* experiment demonstrated that 0.5, 1, 2, and 5 BT had the ability to scavenge free radicals. Therefore, we assumed that drinking BT might have the potential for enhancing antioxidant capacity and scavenging free radicals in organisms.

Subsequently, our data revealed that the plasma from 5 BT drinking group possessed upregulated free radicals scavenging activity. Moreover, this result was contributed by the elevated the glutathione peroxidase (GPx) activity, which was one of the antioxidant enzymes in the modulation of oxidative stress and ROS, especially hydrogen peroxide [23]. These results proved the beneficial and systemic effects of drinking BT, and demonstrated the biological availability of BT that contains water-soluble forms of silicon. Nonetheless, it still requires further researches to confirm the underlying molecular mechanism of these phenomena.

Based on previous researches, water-soluble forms of silicon have more biological availability. Excess amount of water-soluble forms of silicon can be eliminated through kidney within 4–8 hours after consumption, which are unlikely to cause excessive accumulation in healthy individuals [24]. Moreover, average daily dietary intake of silicon is 20–50 mg in populations with western diets and 140–200 mg in Asian population with higher intakes of plant-based foods [25]. However, accurate evidences of oral toxicity in animals or humans are absent, safe upper levels for humans have been recommended with 750–1,750 mg/day [26]. The maximum concentration of silicon in BT is approximately 37 mg/L. Rat average daily water intake is approximately 10 ml/ 100 g body weight, which represents 0.37 mg/100 g body weight of silicon intake through drinking BT. Rat average daily intake of standard rodent feed, containing 0.32 mg silicon/g feed, is approximately 5 g/100g body weight, which represents 1.6 mg/100 g body weight of silicon intake from standard rodent feed. Therefore, the administrations of BT in our study are relatively safe.

Our results confirmed that drinking all types of BT for four weeks did not affect the body weight changes. It was preliminarily confirmed that all types of BT had no adverse effects on animal growth. In addition, the results confirmed that all kinds of BT reduced defecation, which were not related to the weight of animals and the moisture content of the feces. The further researches are needed to elaborate this phenomenon. In regard to gastrointestinal motility, only 5 BT can significantly reduce intestinal transit, which implicated the potential for treating diseases with abnormal peristalsis such as diarrhea.

The gastric ulcer induced by pylorus ligation is mainly resulted from excessive secretions of gastric acid [27]. Excessive gastric acid secretion also stimulates pepsin release and consequently up-regulates the self-digestion of gastric mucosa [28]. Based on our results, it was confirmed that volume of gastric juice was significantly increased after pylorus ligation surgery, whereas this phenomenon was attenuated in 5BT group, which implicated the potential of BT for applying to diseases that caused by excessive secretion of gastric acid.

Damaged gastric mucosa activates the inflammatory process, which increases inflammatory mediators including TNF-α, IL-1β, and IL-6. These cytokines can stimulate neutrophil infiltration and epithelial cell apoptosis, reduce gastric microcirculation around the ulcer region and delay gastric ulcer healing [17]. In present study, 5BT can significantly inhibit the increase of free radicals, erythrocyte extravasation, and gastric mucosa injury induced by accumulation of gastric juice, which contributed to the elevated 4-HNE, one of the lipid peroxidation markers [29], and IL-1β, one of the pro-inflammatory cytokines [30]. However, drinking BT appeared to have no effects on pylorus ligation-induced elevated 3-NT, a marker for reactive nitrogen species [31]. According to current results, drinking 5 BT may not only attenuate the excessive secretion of gastric acid, but also confer gastroprotective effect via reducing gastric mucosal hemorrhage, suppressing pro-inflammatory cytokines from infiltrating leukocyte, upregulating systemic GPx activity, and consequently depressing lipid peroxidation within gastric tissue.

Previous scientific evidences have demonstrated that gut microbiota were profoundly associated with nutrition, health, diseases, and gastrointestinal functions via various metabolites,

including bile acids, amino acids, and short-chain fatty acids. In addition, previous research indicated that the variations in gut microbiota among individuals were highly associated with age, stress, probiotics, prebiotics, and dietary patterns [32], which supported the possibility of gut microbiota modulation through silicon-containing water intake. Present results confirmed that drinking some concentrations of BT increased the relative abundance of *Ruminococcaceae* UCG-005 and *Prevotellaceae* NK3B31 group. *Ruminococcaceae* spp. account for a relatively abundant part of the human gut microbiota and affect several metabolic functions by the production of butyric acid, which may protect healthy subjects from chronic intestinal inflammation [33]. *Prevotellaceae* NK3B31 group can produce short-chain fatty acids and may promote anti-inflammatory responses. Besides, abundance of *Prevotellaceae* NK3B31 group is negatively associated with dextran sulfate sodium induced colitis [34].

On the other hand, some concentrations of BT decreased harmful flora such as *Mucispirillum* and *Rodentibacter*. Previous studies showed that *Mucispirillum* is positively associated with the increased plasma level of lipopolysaccharide, and is assumed to be microbial marker of colitis [35]. *Rodentibacter* spp. are considered as opportunistic pathogens of laboratory rodents. *Rodentibacter* spp. are responsible for the prevalent bacterial infections in laboratory rodents by producing diseases along with other primary pathogens, such as respiratory tract and urogenital tract infection [36].

Moreover, our results indicated that some concentrations of BT increased the abundance of beneficial species such as *Weissella paramesenteroides*, *Lactobacillus reuteri*, *Lactobacillus murinus*, whereas some concentrations of BT decreased the abundance of harmful species such as *Staphylococcus aureus*. According to the published evidence, *Weissella paramesenteroides* has been considered as probiotics because of its antimicrobial activity through the production of bacteriocins such as weissellin A [37]. *Lactobacillus reuteri* is considered as probiotics because of its ability to inhibit the growth of the other potential pathogens such as *Salmonella typhimurium* by secreting antibiotic substances such as reuterin. *Lactobacillus reuteri* can also improve many conditions including diarrheal disease, infantile colic, eczema, and *Helicobacter pylori* infection [38]. *Lactobacillus murinus can* enhance intestinal barrier function, reduce the translocation of bacterial products, reduce systemic inflammatory marker in mice, promote regulatory T cell and inhibit the development of dextran sulfate sodium-induced colitis [39]. Additionally, *Staphylococcus aureus* is a well-known opportunistic pathogen, which causes a variety of disease in humans and animals, such as psuedomembranous colitis, microbiota disruption of human colon and the antibiotic associated diarrhea during antibiotic treatment [40].

On the basis of our results, administrations of BT can partially increase beneficial microbes and decrease harmful microbes, which implicate the effects of BT on gastrointestinal function regulation and the potential of BT as a prebiotic. Recent studies also indicated that regulation of gut microbiota through dietary interventions may be a strategy for prevention or treatment of gastrointestinal diseases, such as inflammatory bowel disease [41]. To the best of our knowledge, this is currently the first research that explore the possible influences of ingesting water-soluble form of silicon on the variation of gut microbiota. Our evidences preliminarily support the potential of BT against gastrointestinal diseases through microbiota modulation. Nonetheless, thoroughgoing studies are still required to clarify the specific role of silicon in the metabolism of gut bacteria, the metabolites of gut microbiota, and the interactions or cross-species communications among beneficial and harmful flora that are affected by BT administration.

## Conclusion

In summary, our preclinical study examined the multiple effects of drinking silicon-containing (BT) water on the rodents. Our prepared BT demonstrated the *in vitro* free radical scavenging

activity. We confirmed the effects of drinking BT on gastrointestinal motility including reduced intestinal transit rate and reduced defecation. Administrations of BT showed significant gastroprotective effects against pylorus ligation induced gastric damage, oxidative stress and inflammation possibly through GPx-related systemic antioxidant activity. We also verified that drinking BT increased some beneficial bacteria and decreased some harmful bacteria, indicating the favorable effects of BT on gut microbiota modulation. For the first time, this research evidenced the beneficial effects of BT on gastrointestinal function and gut microbiota, which demonstrated the therapeutic potential of BT for the prevention or treatment of gastrointestinal disorders. Nevertheless, further studies are still required to identify the exact underlying mechanisms of BT consumption against gastrointestinal disorders.

## Acknowledgments

We appreciated Ming-Chih Chen and Ruey-Ling Deng from Bestec Biotechnology Co., Ltd in Taiwan for providing the equipment and manufacturing BT water.

We appreciated BIOTOOLS Co., Ltd in Taiwan for analyzing DNA sequencing data.

## Author Contributions

**Conceptualization:** Wei-Yi Wu, Jyh-Chin Yang, Chiang-Ting Chien.

**Data curation:** Wei-Yi Wu, Pei-Li Chou, Chiang-Ting Chien.

**Formal analysis:** Wei-Yi Wu, Pei-Li Chou, Chiang-Ting Chien.

**Funding acquisition:** Jyh-Chin Yang, Chiang-Ting Chien.

**Investigation:** Wei-Yi Wu, Pei-Li Chou.

**Methodology:** Jyh-Chin Yang, Chiang-Ting Chien.

**Project administration:** Jyh-Chin Yang, Chiang-Ting Chien.

**Supervision:** Jyh-Chin Yang, Chiang-Ting Chien.

**Validation:** Jyh-Chin Yang, Chiang-Ting Chien.

**Visualization:** Wei-Yi Wu, Pei-Li Chou, Chiang-Ting Chien.

**Writing – original draft:** Wei-Yi Wu, Pei-Li Chou.

**Writing – review & editing:** Jyh-Chin Yang, Chiang-Ting Chien.

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
