## [Decision Letter · Decision Letter 0]

9 Dec 2020

PONE-D-20-28907

Silicon-containing water intake confers antioxidant effect, gastrointestinal protection, and gut microbiota modulation in the rodents

PLOS ONE

Dear Dr. Chien,

Thank you for submitting your manuscript to PLOS ONE. After careful consideration, we feel that it has merit but does not fully meet PLOS ONE’s publication criteria as it currently stands. Therefore, we invite you to submit a revised version of the manuscript that addresses the points raised during the review process.

After careful reading and considering the reviewer comments, I feel that authors need serious effort to modify the manuscript before further consideration. Authors must effort to substantiate the mechanism behind their claims. The authors did not focus on the systemic effect/bioavailability of Si in this study, which is a major concern in this study. The histology figures must be clearly demonstrated (arrows may serve to indicate) and histo-quantification must be included. ROS CL image/raw data must be shown. In addition, authors must suitably address all the points raised by the reviewers and the instructions given by them.

We look forward to receiving your revised manuscript.

Kind regards,

Saikat Dewanjee

Academic Editor

PLOS ONE

Journal Requirements:

"We appreciated BIOTOOLS Co., Ltd in Taiwan for supporting analysis of DNA sequencing data."

"This work was supported by grant from Ministry of Science and Technology in Taiwan (MOST-107-2218-E-003-001).

The funder had no role in study design, data collection and analysis, decision to publish, or preparation of the manuscript."

Additionally, because some of your funding information pertains to commercial funding, we ask you to provide an updated Competing Interests statement, declaring all sources of commercial funding.

In your Competing Interests statement, please confirm that your commercial funding does not alter your adherence to PLOS ONE Editorial policies and criteria by including the following statement: "This does not alter our adherence to PLOS ONE policies on sharing data and materials.” as detailed online in our guide for authors  http://journals.plos.org/plosone/s/competing-interests.  If this statement is not true and your adherence to PLOS policies on sharing data and materials is altered, please explain how.

Please include the updated Competing Interests Statement and Funding Statement in your cover letter. We will change the online submission form on your behalf.

We note that one or more of the authors are employed by a commercial company: Bestec Biotechnology Co., LTD,.

3.1. Please provide an amended Funding Statement declaring this commercial affiliation, as well as a statement regarding the Role of Funders in your study. If the funding organization did not play a role in the study design, data collection and analysis, decision to publish, or preparation of the manuscript and only provided financial support in the form of authors' salaries and/or research materials, please review your statements relating to the author contributions, and ensure you have specifically and accurately indicated the role(s) that these authors had in your study. You can update author roles in the Author Contributions section of the online submission form.

3.2. Please also provide an updated Competing Interests Statement declaring this commercial affiliation along with any other relevant declarations relating to employment, consultancy, patents, products in development, or marketed products, etc.  

Reviewers' comments:

Reviewer's Responses to Questions

**Comments to the Author**

1. Is the manuscript technically sound, and do the data support the conclusions?

Reviewer #1: Yes

Reviewer #2: Yes

2. Has the statistical analysis been performed appropriately and rigorously? 

Reviewer #1: Yes

Reviewer #2: Yes

3. Have the authors made all data underlying the findings in their manuscript fully available?

Reviewer #1: Yes

Reviewer #2: Yes

4. Is the manuscript presented in an intelligible fashion and written in standard English?

Reviewer #1: Yes

Reviewer #2: Yes

5. Review Comments to the Author

Reviewer #1: After going through the manuscript, I think the manuscript may be accepted for publication. However, I would like to mention that a large number of rodents have been involved in this study which is not so good. I would be glad if the authors choose alternative study model for this type of study in future or design their experiment in such a manner that cruelity to animals is minimum.

Reviewer #2: The manuscript entitled “Silicon-containing water intake confers antioxidanteffect, gastrointestinalprotection, and gut microbiota modulation in the rodents” is good enough with proper scientific approach. There are several comments that need to be properly addressed before further consideration of the manuscript.

1. In this study, authors produced silicon-containing water (BT) by guiding tap water via different columns (carbon column, ion exchange resin column, filter column, silicon minerals column) and fed this BT water to animals. They claimed presence of silica conferred gastrointestinal protection. Authors should check the bio-availability of silica in gastric tissue.

2. Authors used both wistar rat and C57BL/6 for this study purpose. They performed all of the experiments in mice models except the pyrolus ligation. Authors should clarify this issue.

3. Authors claimed that increased secretion of gastric juice content elevated free radical generation on gastric mucosa. Then how did they correlate the results of fig 5A and 5B?

4. What is the cause of increased erythrocyte extravasation in 0.5BTtreated pyloric ligation rat models compared to the normal pyloric ligation rat models?

5. Author should explore the possible mechanisms of how BT drinking water helps to increase beneficial bacteria and diminish harmful bacteria.

6. PLOS authors have the option to publish the peer review history of their article (what does this mean?). If published, this will include your full peer review and any attached files.

Reviewer #1: No

Reviewer #2: No

---

## [Author Response · Author response to Decision Letter 0]

4 Feb 2021

Comments from editor:

After careful reading and considering the reviewer comments, I feel that authors need serious effort to modify the manuscript before further consideration. Authors must effort to substantiate the mechanism behind their claims. The authors did not focus on the systemic effect/bioavailability of Si in this study, which is a major concern in this study. The histology figures must be clearly demonstrated (arrows may serve to indicate) and histo-quantification must be included. ROS CL image/raw data must be shown.

Reply:

We appreciate your insightful comments. We have added two assays to evaluate the antioxidant activity of rat plasma with four weeks administrations of silicon-containing water (BT). We used luminol chemiluminescence detection to determine the H2O2 scavenging activity of plasma. In addition, we used glutathione peroxidase activity assay to confirm the increased H2O2 scavenging activity of plasma is correlated with elevated glutathione peroxidase activity of plasma. 

We believe these results can prove the bio-availability of BT that contains water-soluble form of silicon. Moreover, we confirmed that drinking BT can elevate the antioxidant capability in rat plasma, which indicate the beneficial and systemic effects after the administrations of BT. 

We have added the descriptions of these experiments in the sections of methods (in page 9, lines 183, 193-200), results (in page 11, lines 267-278), and discussion (in page 19, lines 414-421). Besides, we have rearranged the order of figures. The results of antioxidant activity of rat plasma are added in figure 2 D and E. The descriptions of these figures are added in each figure caption (figure 2 D in page 12, lines 261, 262) (figure 2 E in page 12, line 262-264).

The raw data of ROS CL counts in the experiments of H2O2 scavenging activity of BT, H2O2 scavenging activity of plasma, and CL counts of gastric mucosa are added in figure 2 A, figure 2 C, and figure 5 B, sequentially. The descriptions of these figures are added in each figure caption (figure 2 A in page 12, lines 257, 258) (figure 2 C in page 12, lines 260,261) (figure 5 B in page 15, lines 335, 336).

In the part of histological examination, we have added yellow arrows to indicate the regions of erythrocyte extravasation. Moreover, we have added immunohistochemistry examination including 4-HNE, 3-NT, and IL-1β staining in response to reviewer’s comments. We have performed quantitative analysis of histology by software ImageJ. We have added the descriptions of these experiments in the sections of methods (in page 10, lines 208-218), results (in page 15, lines 320-332), and discussion (in page 21, lines 452-460). Besides, we have added and rearranged the figures of histological examination and histo-quantification results in figure 5 D and E. The descriptions of these figures are added in each figure caption (figure 5 D in page 15, lines 337,338) (figure 5 E in page 16, lines 338-341). 

We are very appreciative of your precious comments.

Comments from reviewer #1:

After going through the manuscript, I think the manuscript may be accepted for publication. However, I would like to mention that a large number of rodents have been involved in this study which is not so good. I would be glad if the authors choose alternative study model for this type of study in future or design their experiment in such a manner that cruelity to animals is minimum.

Reply:

Dear reviewer, we are very appreciative of your support and suggestions. We will apply alternative methods in accordance with principles of replacement and refinement in the future study. First, we will reduce the sample sizes of each group, and meet the minimum sample sizes standard of journal in the meanwhile. Second, we will apply ultrasensitive detection method with less requirement of plasma sample volume to evaluate to antioxidant activity of plasma. 

Subsequently, we will apply alternative methods to examine the gastrointestinal propulsion without sacrificing animals. For example, we will use test meal containing indigestible beads, or feeding of wireless capsules, or magnetic resonance imaging after feeding of test meal, or functional ultrasonography after feeding of test liquid meal.

Furthermore, we will use epithelial cells derived from gastric mucosal tissue to research the detailed and molecular mechanisms of gastroprotective effects from BT without using animals. We will apply cell experiments to explore the antioxidant, anti-inflammatory, and anti-apoptotic effects of BT administration.

We are very appreciative of your suggestions. We will make effort to minimize the use of laboratory animals with these alternative experimental methods in our future study.

Comments from reviewer #2: 

The manuscript entitled “Silicon-containing water intake confers antioxidanteffect, gastrointestinalprotection, and gut microbiota modulation in the rodents” is good enough with proper scientific approach. There are several comments that need to be properly addressed before further consideration of the manuscript.

1. In this study, authors produced silicon-containing water (BT) by guiding tap water via different columns (carbon column, ion exchange resin column, filter column, silicon minerals column) and fed this BT water to animals. They claimed presence of silica conferred gastrointestinal protection. Authors should check the bio-availability of silica in gastric tissue.

Reply:

Thank you for proposing this disadvantage. In order to confirm whether that drinking BT confer the beneficial effects on animals, we added H2O2 scavenging activity assay of plasma and glutathione peroxidase activity assay of plasma in our revised manuscript. According to the results of elevated H2O2 scavenging activity and glutathione peroxidase activity of rat plasma, we proved the bio-availability of silicon-containing water (BT). We concluded that drinking BT can result in beneficial, systemic, and anti-oxidative effects of on rats.

In order to confirm whether BT confer gastrointestinal protection via antioxidant effect, we added immunohistochemistry examination with 4-HNE, 3-NT, and IL-1β staining. We found that drinking BT can depressed the pyloric ligation-induced oxidative damage marker, 4-HNE, in gastric tissue. In addition, we found that drinking BT can also decreased IL-1β, one of the pro-inflammatory markers, which was induced by pyloric ligation, in gastric tissue. Therefore, we confirmed that drinking BT demonstrated the effect of gastrointestinal protection through elevated antioxidant activity.

We revised and rearranged these results in the sections of abstract (in page 2, lines 37, 38, 41, 42), methods (in page 9, lines 183, 193-200; in page 10, lines 208-218), results (in page 13, lines 267-278; in page 15, lines 320-332), discussion (in page 19, lines 414-421; in page 21, lines 452-460), and figures (in figure 2 C-D, figure 5 D and E).

2. Authors used both wistar rat and C57BL/6 for this study purpose. They performed all of the experiments in mice models except the pyrolus ligation. Authors should clarify this issue.

Reply:

Thank you for noting this issue. We found that we didn’t clearly described these information in the manuscript. We added the descriptions in the sections of methods (in page 8, line 176; in page 9, line 183; in page 9, line 194; in page 10, line 204) and results (in page 13, lines 269, 275; in page 14, line 298; in page 16, line 345).

In brief, we used wistar rats in the experiments of H2O2 scavenging activity of plasma, glutathione peroxidase activity assay in plasma, pylorus ligation induced ulcer, and gut microbiota analysis. C57BL/6 mice were used in the experiments of body weight recording, defecation recording, and gastrointestinal motility. 

3. Authors claimed that increased secretion of gastric juice content elevated free radical generation on gastric mucosa. Then how did they correlate the results of fig 5A and 5B?

Reply:

Thank you for noting this problem. We found these results were not fully persuasive for elaborating the decreased gastric juice secretion and decreased free radicals in gastric mucosa that were affected by BT treatment. 

Therefore, as described above in the reply to question 1, we added immunohistochemistry examination to confirm that BT treatment increase antioxidant capability in rat, then inhibits free radicals generation, and subsequently decrease oxidative damage in gastric mucosa. We hope these revisions can provide more scientific evidences and further support the gastroprotective effect of BT administration.

We revised these results in the sections of results (in page 15, lines 322-332), discussion (in page 21, lines452-460), and figures (figure 5).

4. What is the cause of increased erythrocyte extravasation in 0.5BTtreated pyloric ligation rat models compared to the normal pyloric ligation rat models?

Reply:

Thank you for noticing this issue. We found the figure of H&E staining in 0.5 BT group show an obvious situation of erythrocyte extravasation due to a few area of severe erythrocyte extravasation in the central part of our selected figure. 

After the quantitative analysis of histological stains, we ensured the area of erythrocyte extravasation in 0.5 BT group were smaller than the area of control group, which were only subjected to normal pyloric ligation, whereas there were no statistically significant difference.

In order not to present these confusing and misleading figures of H&E staining in control group and 0.5 BT group, we replaced these figures with other selected figures. We revised these data in the section of figures (figure 5 D and E). 

5. Author should explore the possible mechanisms of how BT drinking water helps to increase beneficial bacteria and diminish harmful bacteria.

Reply:

Thank you for proposing this foresighted comment. 

After careful survey, we found one relevant research that indicated the dietary supplementation with silicate mineral caused beneficial effects on gut microbiota in pullets. (Dietary supplementation with the clay mineral palygorskite affects performance and beneficially modulates caecal microbiota in laying pullets. J Appl Microbiol. 2016 Apr;120(4):1033-40.) That research found the abundance of Paraprevotella and Papillibacter were increased with dietary supplementation of silicate mineral, whereas we found no variation of those two bacteria in our current study. However, we used rats with drinking water-soluble form of silicon in current study, so there might be a weak correlation between our study and previous article due to different species of experimental animal and different forms of silicon.

Moreover, we found two researches that revealed that food additives containing nanoparticles of titanium, iron, silver, gold, or silicon caused harmful effects on gastrointestinal system and gut microbiota. (Impacts of foodborne inorganic nanoparticles on the gut microbiota-immune axis: potential consequences for host health. Part Fibre Toxicol. 2020; 17: 19.) (Food additives containing nanoparticles induce gastrotoxicity, hepatotoxicity and alterations in animal behavior: The unknown role of oxidative stress. Food Chem Toxicol. 2020 Dec;146:111814.) However, those researches emphasize on the harm mainly resulted from nanoparticle, which are more irrelevant to our current study.

According to these limited evidences, we believe our current study is the first research that explore the gut microbiota modulation affected by drinking water-soluble form of silicon. We overall analyzed the variations of beneficial and harmful gut flora that were modulated by several concentrations of silicon-containing water (BT) in current study. Nevertheless, with little evidence, we are currently unable to predict the role of silicon-containing water in the metabolism or the interactions of each bacteria. We will need more extended experiments to discover the specific mechanisms of silicon-containing water (BT) on gut microbiota modulation in our future researches. We added these descriptions of our disadvantages in the section of discussion (in page 23, lines 503-509). 

We genuinely thank you for proposing these disadvantages of our study and providing these constructive suggestions. We sincerely hope that you will give positive consideration to our manuscript.

---

## [Decision Letter · Decision Letter 1]

1 Mar 2021

Silicon-containing water intake confers antioxidant effect, gastrointestinal protection, and gut microbiota modulation in the rodents

PONE-D-20-28907R1

Dear Dr. Chien,

We’re pleased to inform you that your manuscript has been judged scientifically suitable for publication and will be formally accepted for publication once it meets all outstanding technical requirements.

Kind regards,

Saikat Dewanjee

Academic Editor

PLOS ONE

Additional Editor Comments (optional):

Reviewers' comments:

Reviewer's Responses to Questions

**Comments to the Author**

1. If the authors have adequately addressed your comments raised in a previous round of review and you feel that this manuscript is now acceptable for publication, you may indicate that here to bypass the “Comments to the Author” section, enter your conflict of interest statement in the “Confidential to Editor” section, and submit your "Accept" recommendation.

Reviewer #1: All comments have been addressed

Reviewer #2: All comments have been addressed

2. Is the manuscript technically sound, and do the data support the conclusions?

Reviewer #1: Yes

Reviewer #2: Yes

3. Has the statistical analysis been performed appropriately and rigorously? 

Reviewer #1: Yes

Reviewer #2: (No Response)

4. Have the authors made all data underlying the findings in their manuscript fully available?

Reviewer #1: Yes

Reviewer #2: (No Response)

5. Is the manuscript presented in an intelligible fashion and written in standard English?

Reviewer #1: Yes

Reviewer #2: Yes

6. Review Comments to the Author

Reviewer #1: (No Response)

Reviewer #2: (No Response)

7. PLOS authors have the option to publish the peer review history of their article (what does this mean?). If published, this will include your full peer review and any attached files.

Reviewer #1: No

Reviewer #2: No

---

## [Editor Report · Acceptance letter]

19 Mar 2021

PONE-D-20-28907R1 

Silicon-containing water intake confers antioxidant effect, gastrointestinal protection, and gut microbiota modulation in the rodents 

Dear Dr. Chien:

I'm pleased to inform you that your manuscript has been deemed suitable for publication in PLOS ONE. Congratulations! Your manuscript is now with our production department. 

Kind regards, 

on behalf of

Dr Saikat Dewanjee 

Academic Editor

PLOS ONE